# Artificial Neural Networks with Machine Learning Design for a Polyphasic Encoder

**DOI:** 10.3390/s23208347

**Published:** 2023-10-10

**Authors:** Sergio Alvarez-Rodríguez, Francisco G. Peña-Lecona

**Affiliations:** Laboratorio de Fotónica y Materiales, CU-Lagos (Centro Universitario de los Lagos), Universidad de Guadalajara, Lagos de Moreno 47460, Mexico; sergio.alvarez@lagos.udg.mx

**Keywords:** artificial neural networks, machine learning, optical encoder

## Abstract

Artificial neural networks are a powerful tool for managing data that are difficult to process and interpret. This article presents the design and implementation of backpropagated multilayer artificial neural networks, structured with a vector input, hidden layers, and an output node, for information processing generated by an optical encoder based on the polarization of light. A machine learning technique is proposed to train the neural networks such that the system can predict with remarkable accuracy the angular position in which the rotating element of the neuro-encoder is located based on information provided by light’s phase-shifting arrangements. The proposed neural designs show excellent performance in small angular intervals, and a methodology was proposed to avoid losing this remarkable characteristic in measurements from 0 to 180° and even up to 360°. The neuro-encoder was implemented in the simulation stage to obtain performance results, where the main evaluation metric employed to assess the performance is the total error. This proposal can be useful to improve the capabilities of resolvers or other polyphasic sensors used to obtain outstanding precision and accurate data, even when working under hard and noisy industrial conditions.

## 1. Introduction

There are several types of rotative encoders, each with its own advantages and disadvantages. Here are some of them:

1. Optical encoders: Some of these encoders use a disk with slots that interrupts the light emitted by an LED or laser to generate digital signals that indicate position. Optical encoders have high resolution, are precise, and can be used in high-speed applications. However, they are susceptible to electromagnetic interference and may require careful alignment. Other encoders of this type use optical principles such as polarization of light and interference patterns.

2. Magnetic encoders: These encoders use a Hall effect sensor to detect changes in a magnetic field generated by a rotating magnet. Magnetic encoders are resistant to electromagnetic interference and can be used in high-speed and extreme-temperature applications. However, their resolution is limited and may require careful calibration.

3. Inductive encoders: These encoders use coils and magnetic cores to measure position. Inductive encoders have high resolution and are resistant to electromagnetic interference. However, they are less precise than optical and magnetic encoders and are limited to lower speeds.

4. Capacitive encoders: These encoders measure the distance between two electrically charged parallel plates. Capacitive encoders have high resolution and are resistant to electromagnetic interference. However, their use is limited to lower speeds, and they may be sensitive to humidity.

As we said, there are some types of encoders that leverage the polarization properties of light to measure physical variables, such as the positioning of movable elements. Considering the state-of-the-art, the following works on this subject can be found:

In [1], a straightforward encoder design principle is presented, detecting light intensity using a photo-emitter and a photo-detector placed on opposite sides of two polarized filters.

In [2], polarized light is directed towards multiple fixed analyzers and light detectors, which output an electrical signal to a phase processor.

In [3], an angular position is detected by utilizing amplitude and quadrant information.

In [4], polarization difference imaging techniques are employed to calculate the orientation of a rotatable member.

In [5], an encoder is constructed with a multi-channel reception module to generate two signals proportional to the light intensity of the beam from the polarizers.

In [6], a device is presented for detecting a reference position signal and calculating the relative rotation angle between fixed and movable units.

Additional information on this topic can be found in [7,8,9,10,11], where different polarization techniques are also used to design optical sensors.

Some of these kind of sensors also use neural networks as auxiliary components to achieve accuracy in their measurements:

In [12], an optical encoder is designed using convolutional neural networks to detect a robot’s position.

Additionally, in [13], a nonlinear multilayer optical neural network encoder for image sensing is reported.

The closest antecedents can be found in the previous work [14] and in the patent [15], where Alvarez-Rodríguez and Alcalá Ochoa proposed a novel optical encoder based on a phase-shifting algorithm that utilizes the polarization properties of light. However, for devices like this one, the mathematical approach proposed to predict the positional angle of the rotary element cannot be accurate due to multiple factors, e.g., imperfections in the geometry of encoder phases, optical aberrations, nonlinearities in the behavior of photoreceptors, and electromechanical system noise. Therefore, the incorporation of Artificial Intelligence becomes essential for enhancing the reliability of such devices in an industrial context, which is the motif of the present work.

Furthermore, deep learning in conventional neural networks, with multiple processing layers to learn (see [16,17]), has proven to be particularly effective in computer vision and also in a wide range of applications related to the processing of complex and unstructured data. One of the reasons deep learning has been so successful is its ability to automatically learn useful features and representations from data. This has led to significant advances in artificial intelligence and the automation of tasks that previously required detailed manual programming.


*Thus, the main contribution of this work is focused on the design and implementation of conventional artificial neural networks to address information provided by polyphasic sensors, especially the one presented in the next section, utilizing deep learning strategies (i.e., multilayer neural networks), to obtain a high-precision neuro-encoder, which detects the angular position of the rotary element with a negligible margin of error in small angular intervals but is also endowed with some strategies to extend measurement ranges. Likewise, a simple machine learning strategy is proposed to train the neural networks for any angular interval.*


The remainder of the document is organized as follows: Section 2 establishes the problem that is intended to be solved with this proposal; Section 3 is devoted to presenting the neural designs to address the polyphasic signals along with the training algorithm, the machine learning strategy, and an implementation example; Section 4 is focused on the performance results made via simulations for the neuro-encoder; Section 5 is devoted to suggesting the strategy so that the encoder can work in the full range from 0 to 360° without losing its characteristic precision; finally, in Section 6, the main conclusions are given.

## 2. Problem Setting

This work is devoted to designing and implementing Artificial Neural Networks (ANNs) using a machine learning technique to predict the angular position of rotary elements using the device reported in [14].

However, some changes have been made to the original idea of [14] in order to make it compatible with the Neural Network design.

In Figure 1, we have an exploded view of the device where each of its constituent components can be appreciated. In this figure, we have a base that has five holes to accommodate the five photoreceptors whose information constitutes the five inputs of the proposed neural network. The base also has four seats to house the analyzers, which means that the central photoreceptor is without an analyzer. On top of the base, a bearing is also seated, which allows the rotation of a polarizer since the latter is attached to the rotating element of the bearing. A housing unit solidly connects the base to the fixed element of the bearing. Finally, a transparent cover attached to the rotating element allows the entry of light from the outside. As an additional element, the cover can contain an arrow indicating the reading of the angular position, which must coincide with the output of the neural network.

In Figure 2, the main elements of the basic idea of the sensor are shown. Both the photoreceptors and the analyzers within their bases, and the rotating polarizer in its position are displayed.

In Figure 3, we have the assembly of all the constituent elements of the sensor.

To explain the operation of the encoder, we will begin by saying that when a beam of light passes through two consecutive polarizers, one of them fixed and the other with rotary movement (the fixed one called “polarizer” and the rotary one called “analyzer”), the resulting light intensity I(θ) is quantified by the Malus’s law
(1)$$I\left(\theta \right)=I_{0}\;\cos^{2}\left(\theta \right)$$
where $I_{0}$ represents the intensity of light of reference, i.e., the intensity of light before any polarization, and θ is the angular position of the polarizer mounted on the rotary element of the encoder (see [18,19]). Nevertheless, such an arrangement allows us to measure an angle in the interval 0≤θ≤π/2 and to solve this major drawback; the studied sensory device consists of four phases which follow Malus’s law, with phase shift of π/4 as depicted in Figure 4. It should be clarified that the red grids are only a representation for didactic purposes of the polarization axes since they are not perceived by the naked eye.

In this way, the light intensities coming from the reading of each photoreceptor will allow the identification of the angle generated by the polarizer, as depicted in Figure 5, whose value is in the interval 0≤θ≤2π.

According to the equations, relative to the phase shift polarization, in theory, the angle θ can be obtained by
(2)$$\theta =\frac{1}{2}\tan^{-1}\left[\frac{I_{4}-I_{2}}{I_{1}-I_{3}}\right]$$
where the intensity of light for each of the four phases is modeled by
(3)$$I_{1}\left(\theta \right)=\frac{1}{2}I_{0}+\frac{1}{2}I_{0}\;\cos\left(2\theta \right)$$
(4)$$I_{2}\left(\theta \right)=\frac{1}{2}I_{0}-\frac{1}{2}I_{0}\;\sin\left(2\theta \right)$$
(5)$$I_{3}\left(\theta \right)=\frac{1}{2}I_{0}-\frac{1}{2}I_{0}\;\cos\left(2\theta \right)$$
(6)$$I_{4}\left(\theta \right)=\frac{1}{2}I_{0}+\frac{1}{2}I_{0}\;\sin\left(2\theta \right)$$

However, the use of the algorithm shown by the above equations only works in a laboratory environment since the following conditions in the encoder’s manufacturing must be strictly met:

1. The phase shift angles must be exactly 45 degrees for 4 phases, 120 degrees for 3 phases, etc.

2. Both polarizers and analyzers must be free of optical aberrations and must be manufactured perfectly.

3. The photoreceptors must present linear readings according to the intensities marked by Malus’s law and must be free from phenomena of saturation and hysteresis.

4. The device must work free from disturbances and noise.

Then, a math-based algorithm would provide a reading with zero error as long as the device’s manufacturing was absolutely perfect and there was no electromechanical noise around it.

Obviously, for heavy-duty work conditions in the industry, these conditions are practically impossible to meet. Therefore, a method based on artificial intelligence is proposed to obtain the reading of the encoder’s angular aperture.

Summarizing, according to the polyphasic encoder depicted by Figure 1, Figure 2, Figure 3, Figure 4 and Figure 5, the problem to solve is focused on designing and implementing adequate ANNs endowed with a machine learning technique, with an input vector of five intensities of light and one output to predict the angular position, such that none of the four conditions enumerated above are necessary to fulfill in order to obtain an excellent encoder performance.

This work presents a multidisciplinary approach, where several branches of knowledge are involved, as is suggested in [20], to solve the problem that has arisen.

## 3. Neural Networks Design

This study presents several supervised deep learning neural networks, in the sense presented in [21], instead of a single one, with the purpose of observing the behavior of each of them and obtaining conclusions about the dimensions and characteristics that a neural system oriented to perform the proposed task should have.

Thus, a set of four neural networks are being developed to perform a regression task, where the input of each is a vector with the values of five light intensities and the output is a predicted angle, as is shown in Figure 6 where the hidden neural layers are the ones that will process the input information and transform the data in a way that can be used to generate a good angle prediction. The number of hidden layers and the number of neurons in each of them is a hyperparameter that must be determined through experimentation and fine-tuning.

Figure 4 and Figure 5 show the four phases from which the intensities of light $I_{1},I_{2},I_{3},I_{4}$ originate, along with the light’s intensity of reference $I_{0}$.

The first NN is a network with its five respective inputs, a single hidden layer with two neurons, and finally, an output node. This is represented in Figure 7.

Figure 8 shows a more complex neural network with its five respective input nodes, two hidden layers of four neurons each, and its respective output node.

Figure 9 shows the most complex network of the last three since it is similar to the previous network, but its first hidden layer has eight neurons instead of four.

In this last Figure 9, not all connections are shown in order to preserve the quality of the figure. However, in all the networks shown, it is assumed that all the nodes in one layer are connected to all the nodes in the next layer and to all the nodes in the previous layer.

Finally, in Figure 10, a NN with 36 neurons is presented, where the first hidden layer has 27 and the second hidden layer has 8 neurons. As in the last NN, all nodes from one layer to the next are connected, such that there are 359 synaptic weights.

For all the NNs, the synaptic weights are represented with $W_{ij}$, where *i* represents the destination node and *j* is the source node.

Concurrently, the activation functions for all the neurons in the network are the constant 1 since after attempting to work with other activation functions, the network does not work correctly. The activation functions that were tested were tanh, sig functions, and combinations of exponentials with sin and cos functions. Thus, the first conclusion is that the neural network for this type of application does not require the introduction of these types of nonlinearities.

Furthermore, the numerical value for each node is given by:(7)$$N_{i}=\sum_{j=1}^{n}X_{j}\;W_{ij}$$
which is graphically exemplified in Figure 11.

### 3.1. Training Algorithm

Deep learning is a subfield of machine learning that relies on deep artificial neural networks to perform pattern recognition and decision-making tasks automatically (see [17,21]). These deep neural networks consist of multiple layers of interconnected nodes, which enable them to learn and represent complex features from data.

To train the network, a labeled dataset containing values of light intensity and corresponding predicted angles will be needed. These data will be used to adjust the weights of the network and improve its ability to make accurate predictions. The goal will be to minimize the error between predicted angles and actual values.

To accomplish this task, the total error must be obtained with
(8)$$\epsilon =X_{R}-X_{OUTPUT}$$
where $X_{R}$ is the real value of the angle, and $X_{OUTPUT}$ is the predicted angle. After this, the total error ϵ must be retro propagated to the NN, multiplying the error corresponding to each node by its respective synaptic weight $W_{ij}$.

When the error has been retro propagated, the synaptic weights must be updated by
(9)$$W_{ij}^{*}=W_{ij}+\delta \;\epsilon_{ij}\;X_{j}$$
where δ is the learning rate.

The updated numerical values for each node $N_{i}^{*}$ are obtained by
(10)$$N_{i}^{*}=\sum_{j}W_{ij}^{*}\;X_{j}.$$

When the output value has been obtained, this process is repeated using many training examples.

Once the neural network has been trained, it can be used to make predictions on new sets of data. The accuracy of the network will depend on the quality of the training data and the architecture of the network.

### 3.2. Training Example

The last Algorithms (Equation 8)–(Equation 10) are now exemplified to train the NN given by Figure 8. However, to train the NNs of Figure 7 and Figure 9, the procedure is the same.

1. The first step is the obtaining of the total output using (Equation 8) in three consecutive stages.

(a) To obtain the value for each neuron at the first hidden layer (1≤i≤4):(11)$$N_{i}=\sum_{j=0}^{4}I_{j}\;W_{ij}$$

(b) To obtain the value for each neuron at the second hidden layer (5≤i≤8):(12)$$N_{i}=\sum_{j=5}^{8}N_{\left(j-4\right)}\;W_{i(j-4)}$$

(c) Finally, to obtain the value for the total output of the NN:(13)$$N_{9}=\sum_{j=5}^{8}N_{j}\;W_{9j}$$

The total output ($N_{9}$), which is the predicted angle, is compared with respect to the expected actual value $X_{R}$, obtaining the total error (the value we wish to minimize), given by: (14)$$\epsilon =X_{R}-N_{9}$$

2. The second step consists of the retro propagation of the total error to each connection of the NN, also in three reversed stages.

(a) To retro propagate the error from $N_{9}$ to the second hidden layer (5≤j≤8):(15)$$\epsilon_{9j}=\epsilon \;W_{9j}$$

(b) To retro propagate the error from the second hidden layer to the first hidden layer (5≤i≤8 and 1≤j≤4):(16)$$\epsilon_{ij}=\epsilon \;W_{ij}$$

(c) Finally, to retro propagate the error from the first hidden layer to the input nodes (1≤i≤4 and 0≤j≤4), Equation (Equation 16) is also used.

3. The third step consists of the updating of the synaptic weights using (Equation 9), (Equation 15), and (Equation 16) in the following forwarding direction.

(a) From the input vector to the first hidden layer (1≤i≤4 and 0≤j≤4):(17)$$W_{ij}^{*}=W_{ij}+\delta \;\epsilon_{ij}\;I_{j}$$

(b) From the first to the second hidden layer (5≤i≤8 and 1≤j≤4):(18)$$W_{ij}^{*}=W_{ij}+\delta \;\epsilon_{ij}\;N_{i-4}$$

(c) Finally, from the second hidden layer to the output (5≤i≤8):(19)$$W_{ij}^{*}=W_{ij}+\delta \;\epsilon_{ij}\;N_{i}$$

4. The last step is the obtaining of the new output, and, in some sense, it is simultaneous with step 3 because the updating of the synaptic weights also requires the update of the last outputs, with the exception of the total output, which must be obtained using (Equation 10).

### 3.3. Machine Learning Strategy

The acquisition of information vectors for training a neural network is a time-consuming and laborious process. For this reason, it is desirable for the machine to have the ability to learn by itself using some strategy.

A mathematically based strategy to use the given Equations (Equation 3)–(Equation 6) to automatically obtain the values of light intensities for each angular value is desired, provided that in the referred equations structural perturbations and system noise represented by $\lambda_{k}$ (where *k* is the number of phases) are modeled and incorporated in the following form:(20)$${}^{*}I_{1}\left(\theta \right)=\frac{1}{2}I_{0}+\frac{1}{2}I_{0}\;\cos\left(2\theta \right)+\lambda_{1}$$
(21)$${}^{*}I_{2}\left(\theta \right)=\frac{1}{2}I_{0}-\frac{1}{2}I_{0}\;\sin\left(2\theta \right)+\lambda_{2}$$
(22)$${}^{*}I_{3}\left(\theta \right)=\frac{1}{2}I_{0}-\frac{1}{2}I_{0}\;\cos\left(2\theta \right)+\lambda_{3}$$
(23)$${}^{*}I_{4}\left(\theta \right)=\frac{1}{2}I_{0}+\frac{1}{2}I_{0}\;\sin\left(2\theta \right)+\lambda_{4}$$
where $\lambda_{k}$ is the term that must be absorbed by the neural network and can be obtained using
(24)$${}^{*}I_{k}\left(\theta \right)-I_{k}\left(\theta \right)$$

To carry out this process, it is necessary to implement a system where the angle value is updated, and with that value, the light intensities are obtained. Subsequently, the network is trained with those values, and this entire process is repeated automatically in a repetitive manner.

In the initial step, an arbitrary value $\theta_{0}$ is assigned to the angle θ, and the given equations are used to calculate the light intensities $I_{1}^{0}$, $I_{2}^{0}$, $I_{3}^{0}$, $I_{4}^{0}$. Then, the angle is incremented/decremented to obtain $\theta_{1}$, leading to the modification of the light intensities to $I_{1}^{1}$, $I_{2}^{1}$, $I_{3}^{1}$, $I_{4}^{1}$, and the synaptic weights are adjusted accordingly.

This process of updating the angle, calculating the new intensities, and modifying synaptic weights is repeated for *n* iterations:(25)$$\theta_{n}\to I_{1}^{n},I_{2}^{n},I_{3}^{n},I_{4}^{n}$$

For a detailed procedure in implementing these concepts, see the explanation of Figure 12 in the next section.

In order to model the function of nonlinearities, noise, and static and dynamic disturbances of a specific device, it is advisable to first carry out the process described manually to obtain the light intensities, and then use the proposed Equation (Equation 24).

The other alternative to using machine learning techniques is to train the network manually, which would increase, in an extreme way, the time and effort devoted to obtaining results and adjustments to the neural network

### 3.4. Implementation Example

This implementation example is performed in Matlab’s Simulink program and involves applying the Equations (Equation 11)–(Equation 19) that describe the neural network in the Figure 8 along with its training, and the machine learning strategy (Equation 20)–(Equation 25).

The same principles and programming techniques from this example are followed to program the other neural networks proposed in this work. Figure 13 shows the root program, which consists of eight subsystems. The first subsystem contains the neural network with five inputs, two hidden layers with four neurons each, and one output neuron.

The second subsystem contains the program for the first training cycle, where the error is backpropagated, synaptic weights are adjusted, and an output is obtained with reduced error.

The third subsystem contains a program that calculates the corresponding light intensities for an angle that has a different value than the initial one. Concurrently, the fourth subsystem contains the training program for this third subsystem, following the same description as for subsystem two.

The fifth and sixth subsystems are a repetition of the third and fourth subsystems. Similarly, the seventh and eighth subsystems are also a repetition of the previous ones, allowing for the addition of the desired number of pairs.

The $W_{1},\cdots ,W_{9}$, represent vectors of synaptic weights for each of the nine neurons in the exemplified network, totaling forty parameters that are updated in each angle update and its corresponding training. This process continues until the end of the sequence, where the values of the synaptic weights are stored in Matlab using the “To Workspace” instruction to initiate new training cycles that progressively reduce the error.

If, for example, we start with an angular value of 17 degrees and program a negative increment of one degree, then subsystem eight will provide us with adapted synaptic weights and corresponding light intensities for 14 degrees. However, prior to this, the network has been trained for values of 15, 16, and 17 degrees. If we repeat this process a sufficient number of times, it is expected that the network will be trained to predict angular measurements close to the 14–17 degree interval when given a vector of light intensities as input.

Both the accuracy and precision, as well as the range of the angle to be predicted, depend on the capacity of the neural network. In other words, a neural network with small hyperparameters and insufficient architecture will be able to predict within small angular intervals with wide margins of error. At the same time, powerful neural networks can predict with high accuracy and precision within the full 360-degree range detectable by the sensor.

Moving on to the next Figure 14, we have the block programming of what is contained in subsystems three, five, and seven of the previous Figure 13. In program of Figure 14, we can see other subsystems, one of them containing the error backpropagated inherited from the previous subsystem, another one containing the calculation of light intensities for new angles, and four others containing the neurons of the first hidden layer. The four summation nodes appearing on the right-hand side represent the four neurons of the second hidden layer. The rightmost summation node represents the output neuron, where the calculation of the total error at that stage is performed. As it can be seen, the synaptic weights are updated before the calculation of each neuron’s output.

In Figure 12, the program within the subsystem “Intensities of light” is shown, which corresponds to the calculation of the light intensities that should correspond to each angular measurement, according to what was explained in the subsection “Machine learning strategies”. Figure 12 also includes disturbances and a white noise generator to simulate real-world conditions.

The input “a” is the angle stored in the program’s working memory, starting with an arbitrary initial value. Concurrently, the constant input “b” is the increment/decrement that will be applied to the angle in each iteration of the training process. By adding/subtracting “b” from “a”, an updated value of “a” is obtained, which will change throughout the machine learning cycle.

If we take away the subsystem of Figure 12 from Figure 14, then subsystems four, six, and eight in Figure 13 are obtained because the exclusive training blocks do not include an update of angle and light intensities. The same can be said for block two in Figure 13; however, the first training block does not inherit previous synaptic weights.

The block program within the “Retro propagation” subsystem is shown in Figure 15, where items 3 and 4 of the “Training example” can be seen in a diagram block.

Finally, Figure 16 shows the interior of a neural subsystem where it can be observed that the only constant value is $I_{0}$. In this case, we are dealing with a neuron in the first hidden layer. As you can see, the synaptic weights converging on the node are multiplied by the light intensities from the encoder. The neurons in the second hidden layer have a similar structure but with four inputs coming from the first hidden layer.

Up to this point, the neural network of Figure 8 has been exemplified using block diagrams and explanatory text, allowing the reader to implement this neural network on their own. Finally, as you may have noticed, for this program to work, two additional Matlab programs in .m format are required. The first program should be executed before running the block program and should contain the parameters “a” and “b” along with their initial light intensity values, the learning rate “RA”, and the initial synaptic hyperparameter. The second .m program should be executed after each run and should contain the updated values of the synaptic weights sent to Matlab using the “To Workspace” instruction.

Now let us show another example with an arrangement made for the NN of Figure 10 endowed with 36 neurons, which corresponds to the program shown by Figure 17, which has a different purpose than the one in the Figure 13. Every “Stage” block of Figure 17 has the same structure of the “Increment” block in Figure 13, and both kinds of structures will be used to obtain simulation results.

Observe that the hyper parameter for this NN is conformed by 359 synaptic weights.

Once the NNs have been trained, the hyper parameters can be stored in a database to be used when they are required. In the Appendix A, Appendix B, Appendix C and Appendix D, the initial synaptic weights, along with the trained ones, are provided for every NN.

Note: The synaptic weights for $I_{0}$ do not appear for the first NN (3 neurons) since, heuristically, it was detected that they did not provide benefits for their good performance.

## 4. Performance Results

The implementation for simulation purposes has been performed for the four networks shown in the Figure 7, Figure 8, Figure 9 and Figure 10, and below performance results are presented for an interval from 13 to 17 degrees, and symmetrically from 17 to 21 degrees, using the corresponding light intensity vector as a system input. For this purpose, each NN has been implemented in two versions, the first one with the structure shown by Figure 13, and the second one with a structure as shown in Figure 17.

The intensities of light for every angle θ were obtained using Equations (Equation 3)–(Equation 6), and taking $I_{0}=50$ as the parameter for all simulation results.

For this first set of results no perturbations $\lambda_{k}$ have been included in (Equation 20)–(Equation 23). Nevertheless, a second study has been performed to investigate the effect of including $\lambda_{1},\lambda_{2},\lambda_{3},\lambda_{4}$ as white noise functions with a peak amplitude of 0.00012 and an average frequency of 650 Hz. Due to the strange behavior of neural networks, which can vary significantly based on their architecture and size, establishing benchmarks for comparison can be a challenging task. Nevertheless, given its specific role within this application, we will endeavor to devise a methodology for conducting meaningful comparisons.

Using the novel machine learning technique expressed by (Equation 25) and illustrated by Figure 13, a performance comparison between NN2, NN3, and NN4 is given in Table 1 (NN1 is not considered because it presents low performance when it is trained for intervals).

The intervals in the first column of the referred table have a lower value, a central value, and an upper one, such that the central value for every interval has been arbitrarily selected as 17 degrees for this study. This central value increases or decreases symmetrically in one, two, three, and four degrees to obtain four intervals.

Remember that every increment block increases/decreases “b” degrees (for this case b = 1) from the previous block (accordingly with Figure 12), such that the performance of the first interval is obtained taking data from the Training 1 block, the second interval takes data from the Training 2 block, while the third interval takes data from the Training 3 block, shown in the mentioned Figure 13.

The table indicates the minimum values obtained for the training at the extremes of every interval, taking into account the better learning rate (LR) heuristically found and used for each case.

Certain remarks can be drawn from Table 1:

1. When the range of the interval is widened, the minimum error increases. Nevertheless, an increase in network capacity helps in reducing error.

2. The neural network of 36 neurons performs a little better than the others; however, the computational cost increases considerably.

3. Strangely, the network of 9 neurons is more efficient than that of 13, which supports the fact that the architecture of the network influences its efficiency.

4. Both the errors and the training rates of the network of 13 neurons are higher than those of the others (for the last three intervals), which supports the idea that both concepts are directly proportional to each other for a neural network (also see that in the first interval, it is reversed).

5. An additional remark obtained from the research process is that every one of the presented Neural Networks is extremely sensitive to small hyper parameter changes since a single change of some initial synaptic weight value can lead to different NN behavior. Nevertheless, those of Table 1 are the better results obtained after many hours of tune and adjustment. Thus, it is concluded that a lot of time is required to tune the NNs and to find their optimal functioning.

Once a comparative analysis between the different neural networks has been carried out, let us review the error that each of them throws when they are trained around a specific region. The angle must be specified by the user assigning the desired angular value “a” and “b = 0” in the Matlab console directly or running beforehand a .m file containing this information, which feeds the subprogram shown in Figure 12, and after running the program depicted in Figure 17.

For this case is selected “a = 17” (which with ±0.5 can be used to detect angles in the range 16.5–17.5). The corresponding intensities of light are automatically obtained by subprogram of Figure 12, and the training process starts with the machine learning technique.

In Table 2, Table 3, Table 4 and Table 5, the first column corresponds to the Learning Rate (LR), and the first row to the training stage from the initial reading (0) to the 9th learning stage.

The initial readings in column 0 were obtained using randomly assigned synaptic weights, while the machine learning technique updated the synaptic weights from the 1st to the 9th stages in an extremely fast process.

Table 2, shows the training performance of the first NN shown in Figure 7, where the best LR is 0.0005 and reaches a minimum error of −1.2$\times \;10^{-4}$. Figure 18, shows graphically the 4th row of Table 2.

In Appendix A, both the initial and the final synaptic weights (after nine training stages using the machine learning technique) are given for the NN with three neurons, excluding the $I_{0}$ connections.

Table 3 shows the training performance of the first NN shown in Figure 8, where the best LR is 0.0001 and reaches a minimum error of −0.533. Figure 19, shows graphically the 5th row of Table 3.

In Appendix B, both the initial and the final synaptic weights are given for the NN with nine neurons, where the initial synaptic weight corresponds to the table’s column 0 and the final weights are those obtained from the nine training stage.

Table 4 shows the training performance of the third NN shown in Figure 9, where the best LR is 0.0001 and reaches an error of 1.72$\times \;10^{-4}$. Figure 20, shows graphically the 5th row of Table 4.

In Appendix C, both the initial and the final synaptic weights are given for the NN with thirteen neurons, where the machine learning technique has been used to obtain the last trained weights (during nine stages).

Table 5, shows the training performance of the third NN shown in Figure 10, where the best LR is 0.001 and reaches an error of −1.1$\times \;10^{-10}$. Figure 20 shows graphically the 3rd row of Table 5.

To observe the evolution of the synaptic weights from their initial to the final values in the training process using the proposed machine learning technique, please see Appendix D.

As mentioned at the beginning of this section, the presented results were obtained using a machine learning model without any disturbance. However, experiments were conducted by introducing white noise to dynamically perturb the light intensity values, resulting in an accuracy approximately 4% lower than the results obtained with the undisturbed system. Clearly, as the dynamic disturbance increases, the accuracy continues to decrease. A more comprehensive study on this matter is proposed as a topic for future research in the context of applications involving polyphasic sensors with perturbed input vectors. It will be hypothesized that static disturbances are completely absorbed by the neural network.

Obtaining the results presented in each of the above tables using manual neural training methods can take several days of hard work per run, while the use of the proposed computer machine learning technique has reduced that time to only a few minutes per run using a standard computer.

## 5. Intervals Management

In Table 6, the full range from 0 to 180${}^{\circ }$ is partitioned in 16 intervals of 11.25${}^{\circ }$ each. In the first column, the angular position is marked, and in columns 2, 3, 4, and 5, the corresponding intensities of light are given (taking $I_{0}=50$). Observe that every interval has a defined and unrepeated relationship between the four intensities of light. For example, the first interval goes from 0 to 11.25${}^{\circ }$, and the relationship between the corresponding intensities of light is:(26)$$\begin{array}{c} I_{1}>I_{4}\geq I_{2}>I_{3}\\ 50\geq I_{1}\geq 48.1\\ 25\geq I_{2}\geq 15.42\\ 0\leq I_{3}\leq 1.9\\ 25\leq I_{4}\leq 34.57 \end{array}$$

Obviously, the beginning of the first interval, or in other words, zero degrees, can be located at any position on the circumference, so it is possible to put any interval (e.g., 16-17-18, 15-17-19, 14-17-20, 13-17-21, and so on) in the center of any interval of Table 6.

Under these considerations, the following methodology to measure angular positions from 0 to 180${}^{\circ }$ with grate precision is suggested:

1. Train the NN with 36 neurons, according to the scheme of Figure 10 for each of the 16 intervals given by Table 6.

2. Store the values of the synaptic weights trained for each of the intervals in a data bank.

3. Use an automatic system to identify the interval in which the reading is found based on the relationships between the values of light intensities.

4. Call the package of synaptic weights that correspond to the identified interval and use it to make the prediction of the angular measure.

Additionally, an index on/off detector can be located in the encoder to detect from 180${}^{\circ }$ to 360${}^{\circ }$, and then a full range neural encoder can be obtained with outstanding accuracy.

## 6. Conclusions

A neuro-encoder that solves the drawbacks caused by mathematically based algorithms to address polyphasic signals has been designed and presented in this work. Unlike math-based algorithms, the deep learning neural network does not require a perfect geometry of encoder phases, optical components free from aberrations, linear behavior of photoreceptors, or a system free from electromechanical noise. This represents a significant reduction in manufacturing costs and makes the encoder suitable for operation under rugged industrial conditions with high reliability. The maximum precision achieved in previous work using Equation (Equation 2) to obtain readings through mathematical methods was 1.39 $\times \;10^{-2}$ under laboratory conditions. In contrast, the angular precision achieved using neural network 4, disregarding ideal conditions, is 1.1 $\times \;10^{-10}$ in the 9th stage of self-training. Therefore, the neural network makes the polyphase encoder 126 million times more accurate, in small measurement intervals, than the math-based algorithm.

Thus, the proposed neural architecture is ideal for predicting, with great accuracy, positions that are within small angular intervals. It performs these predictions in such a way that the precision and accuracy of the neural device are inversely proportional to the length of the angular interval. The characteristics of the light intensities provided by the encoder allow segmenting the 180${}^{\circ }$ of the semi-circumference into 16 intervals, which is very convenient for this type of artificial intelligence applications.

The accuracy, precision, and range of the angle to be predicted are influenced by the capacity of the neural network. A neural network with small hyper parameters and limited architecture may struggle to predict within wide angular intervals and may have larger margins of error. On the other hand, more powerful neural networks, with larger architectures and better training, have the ability to predict with higher accuracy and precision across a wider range of angle measurements. Properly choosing the neural network architecture and optimizing its hyper parameter guarantee achieving optimal performance in angular prediction.

The most notable advantage of using the machine learning technique, exemplified by Figure 12, is the enormous savings in time and effort in tuning synaptic weights for the neural networks. Furthermore, Figure 18, Figure 19, Figure 20 and Figure 21 show the outstanding performance of the proposed math-based self-learning strategy.

Finally, in Figure 22 and Figure 23 we show an example of the behavior of the synaptic weight values of $w_{1}$ from 0 to the 9th training stage, where the robustness of the applied technique can appreciate.

## 7. Future Work

For future work, it is proposed to find and study a neural architecture capable of reading the angular positions of polyphase encoders in the range of 0 to 360 degrees with extraordinary precision, directly, without the need for interval segmentations or constraining readings to regions. Another natural follow-up task is to implement these proposed techniques with other types of encoders and polyphase signal devices, such as resolvers and multi-channel potentiometers.

Another future work can be a deeper study of the math-based machine learning technique, focused on modeling both the static and the dynamic disturbances $\lambda_{k}$ for every kind of polyphasic sensor. 

## Figures and Tables

**Figure 1 sensors-23-08347-f001:**
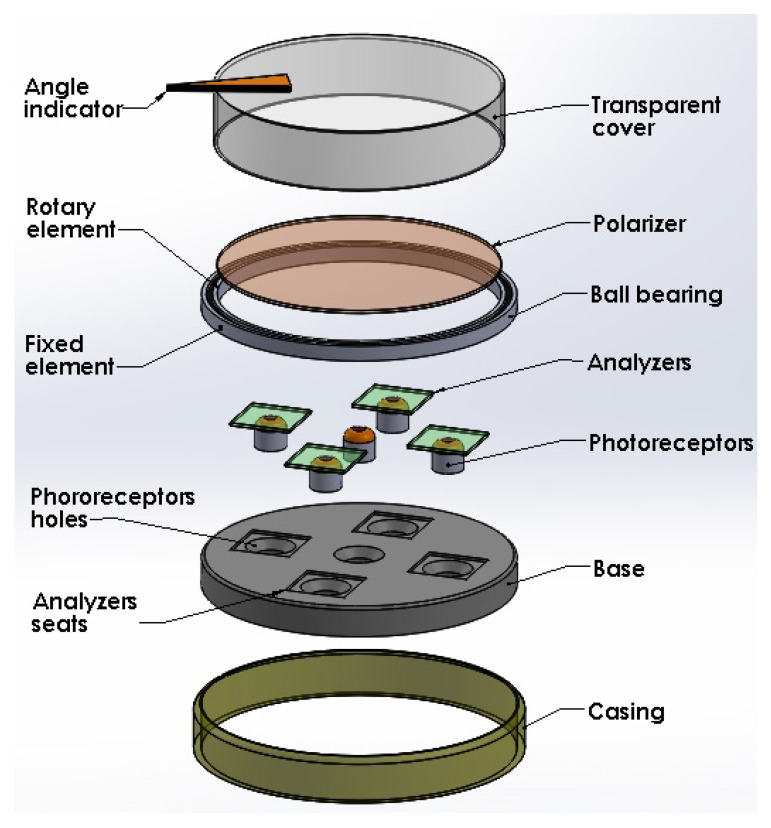
Exploded view of the encoder.

**Figure 2 sensors-23-08347-f002:**
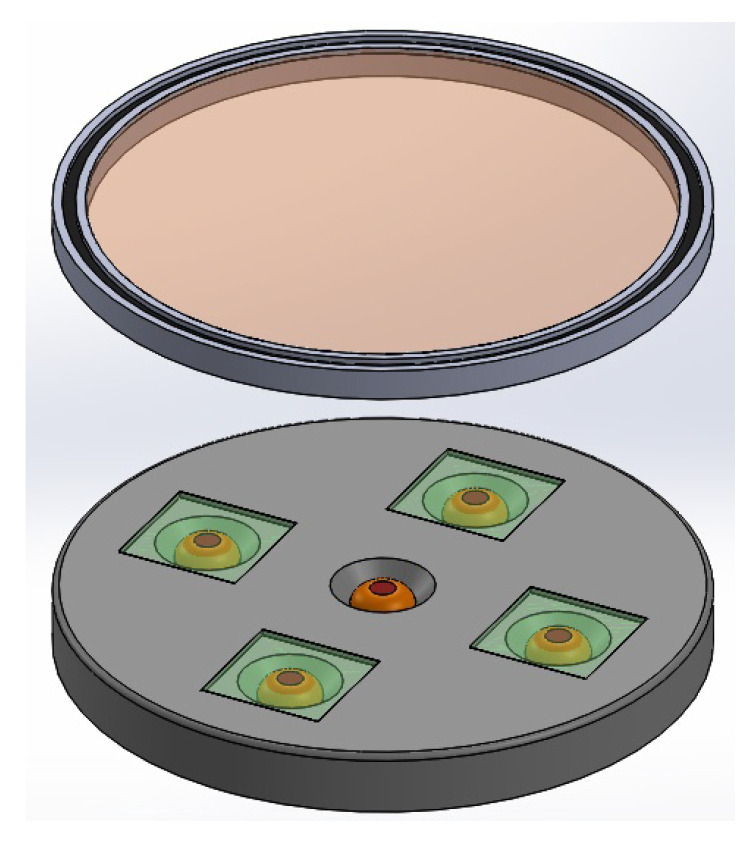
Main elements of the encoder.

**Figure 3 sensors-23-08347-f003:**
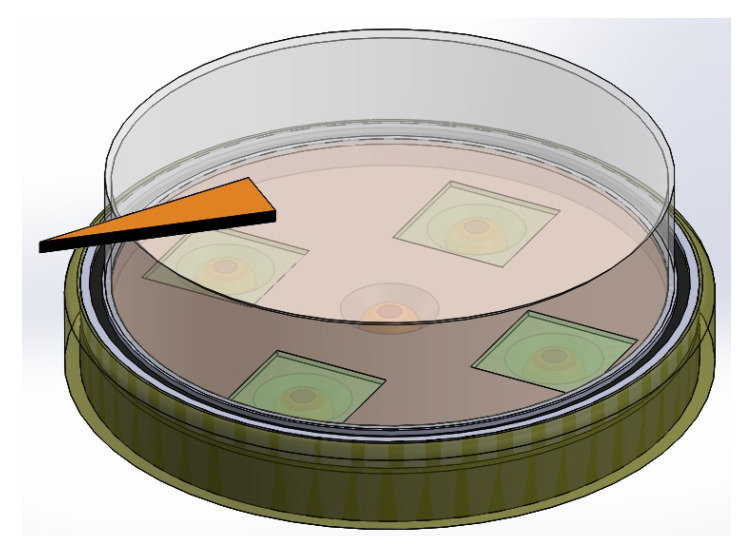
Assembly of the elements shown in Figure 1.

**Figure 4 sensors-23-08347-f004:**
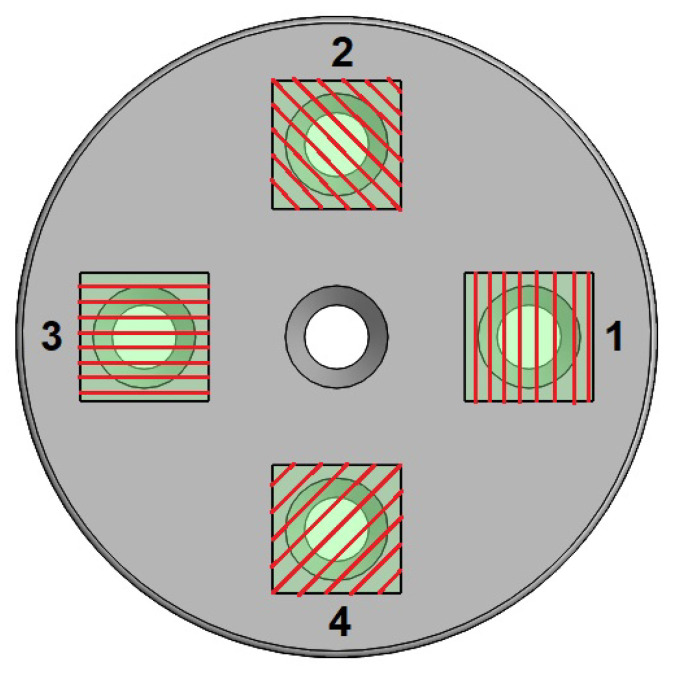
Orientation of the polarizing axes of each analyzer.

**Figure 5 sensors-23-08347-f005:**
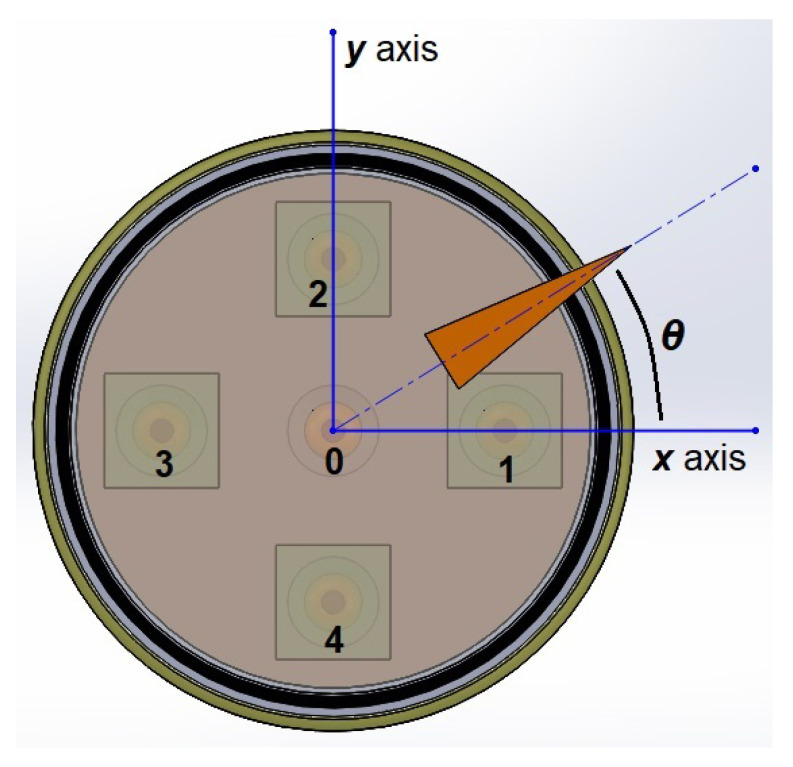
Angular position of the rotary polarizer.

**Figure 6 sensors-23-08347-f006:**
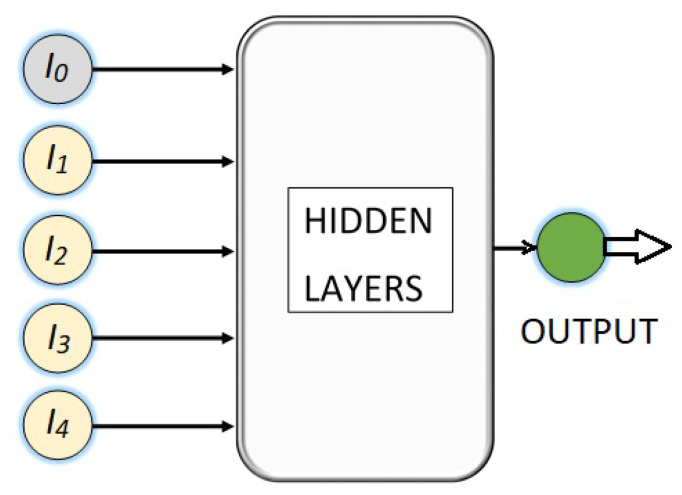
Basic design of the deep NN for the polyphasic encoder.

**Figure 7 sensors-23-08347-f007:**
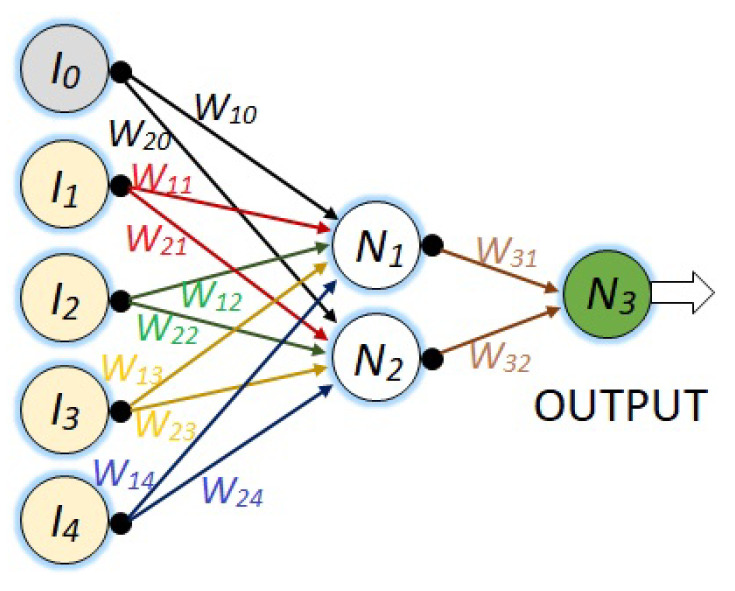
NN with five inputs, one hidden layer of two neurons, and the output neuron.

**Figure 8 sensors-23-08347-f008:**
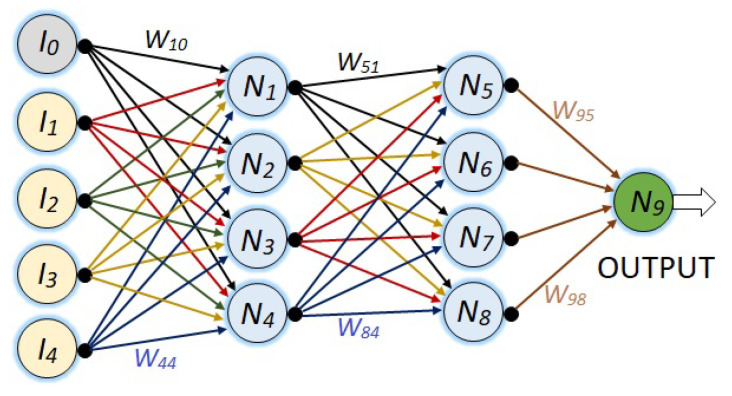
NN with five inputs, two hidden layers of for neurons each, and the output neuron.

**Figure 9 sensors-23-08347-f009:**
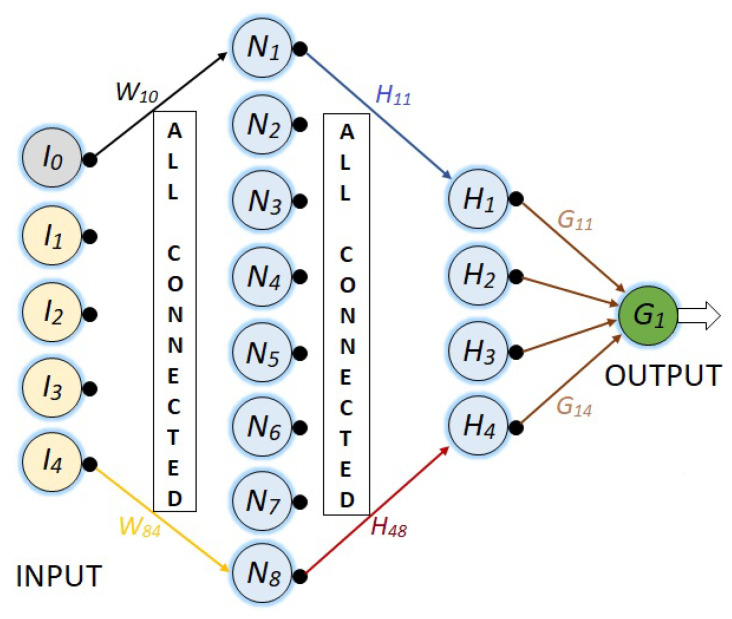
NN with five inputs, two hidden layers: the first one with eight and the second one with four neurons, and the output neuron.

**Figure 10 sensors-23-08347-f010:**
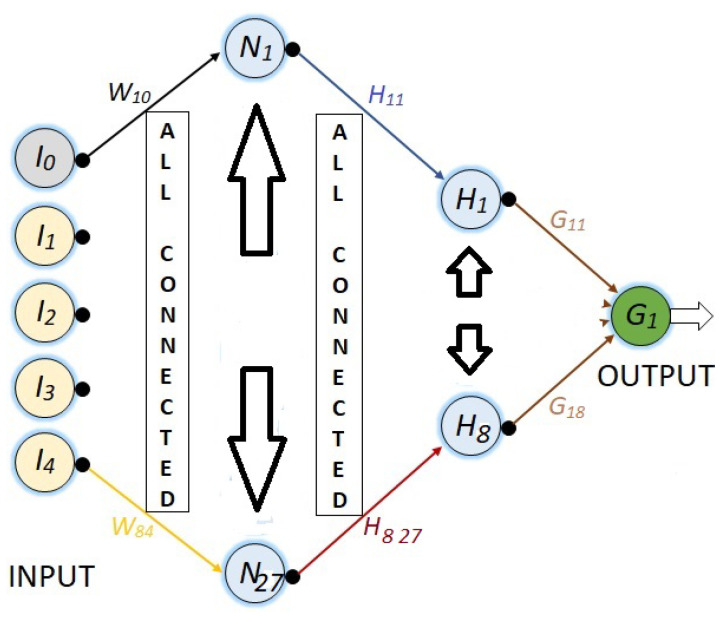
NN with five inputs, two hidden layers: the first one with 27 and the second one with 8 neurons, and the output neuron.

**Figure 11 sensors-23-08347-f011:**
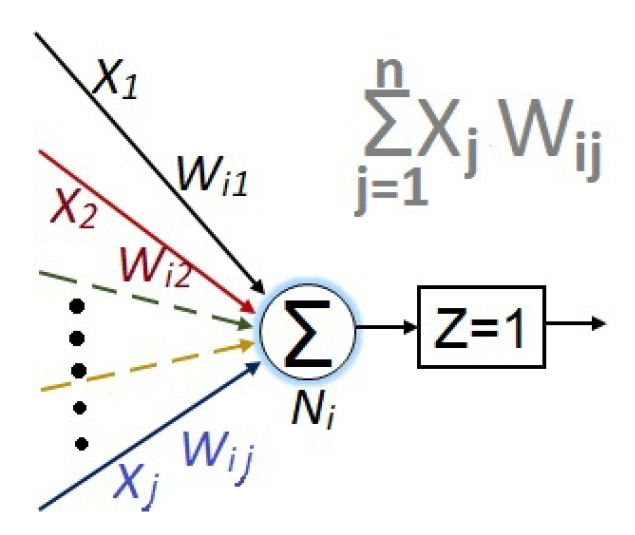
An individual neuron.

**Figure 12 sensors-23-08347-f012:**
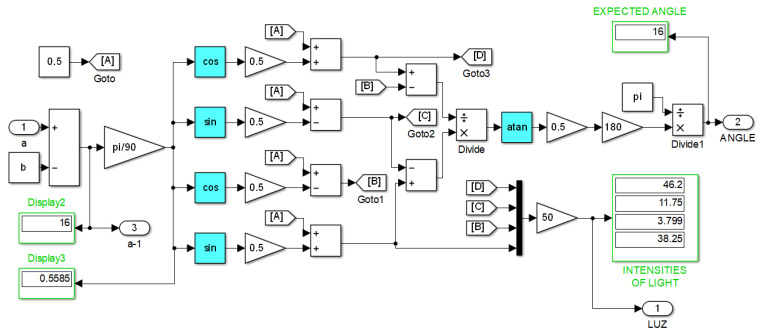
Automatic calculation of angular position along with its corresponding light intensities.

**Figure 13 sensors-23-08347-f013:**
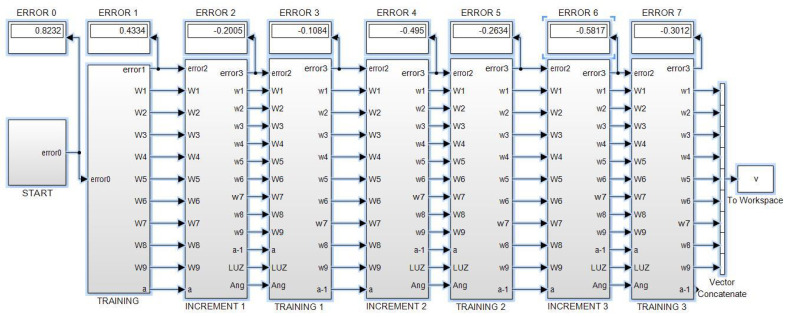
Root program of the machine learning technique.

**Figure 14 sensors-23-08347-f014:**
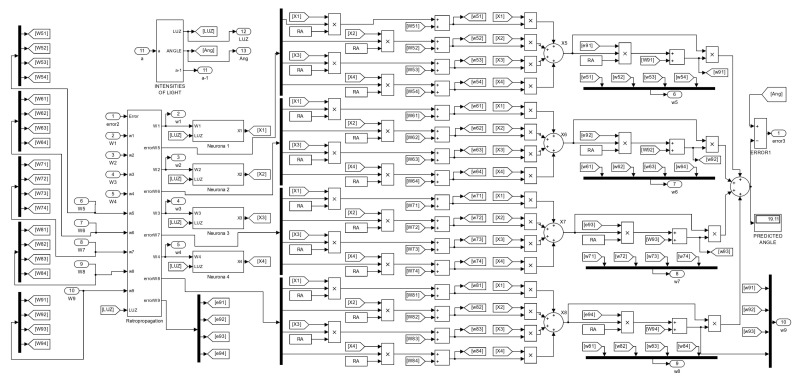
Program inside the third, fifth, and seventh subsystem blocks of Figure 13.

**Figure 15 sensors-23-08347-f015:**
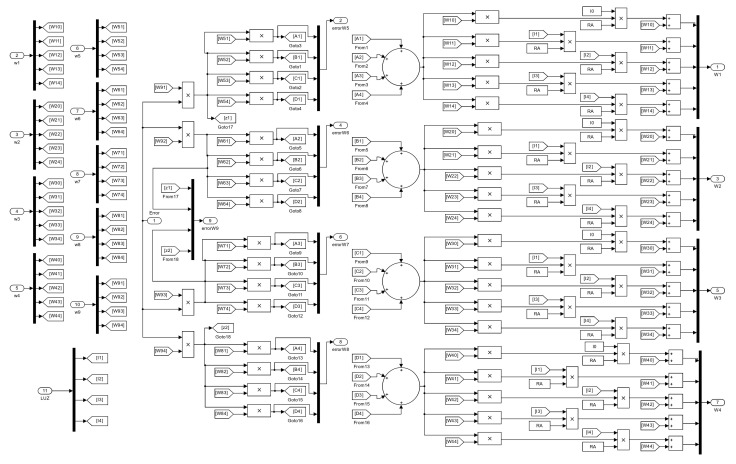
Program for the retro propagation of the error.

**Figure 16 sensors-23-08347-f016:**
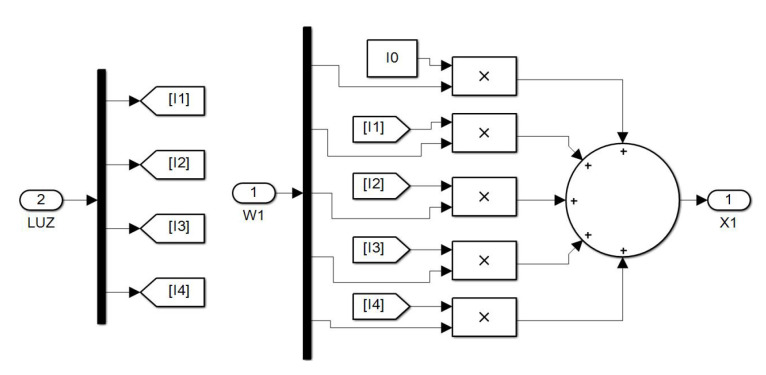
Interior of a neural subsystem.

**Figure 17 sensors-23-08347-f017:**
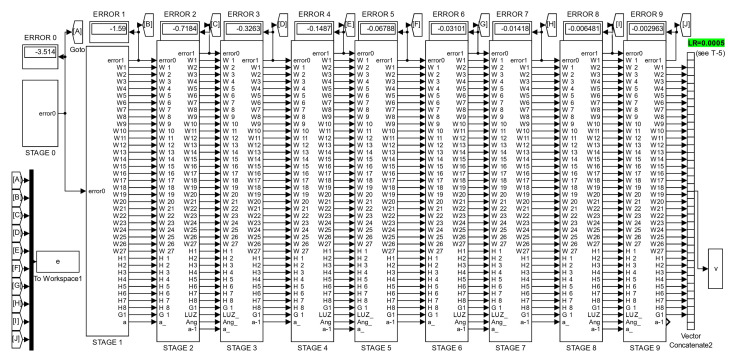
Neural training around an specific angular value.

**Figure 18 sensors-23-08347-f018:**
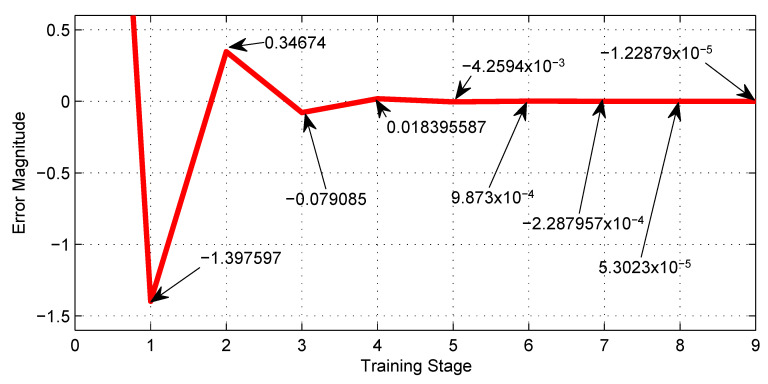
Error behavior of the NN with three neurons given by Figure 7 along the training stages from the initial 0 to the 9th.

**Figure 19 sensors-23-08347-f019:**
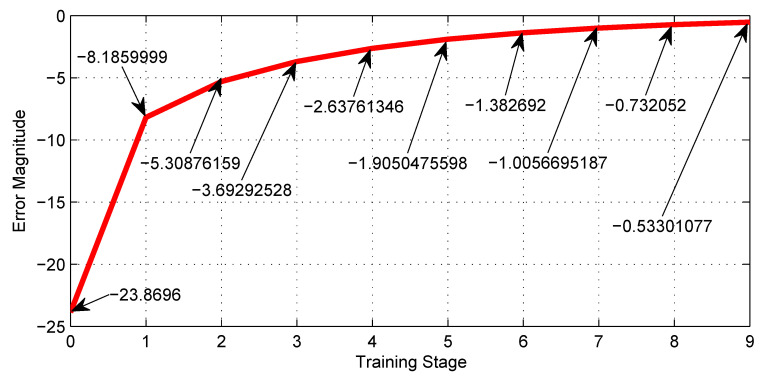
Error behavior of the NN with nine neurons given by Figure 8 along the training stages from the initial 0 to the 9th.

**Figure 20 sensors-23-08347-f020:**
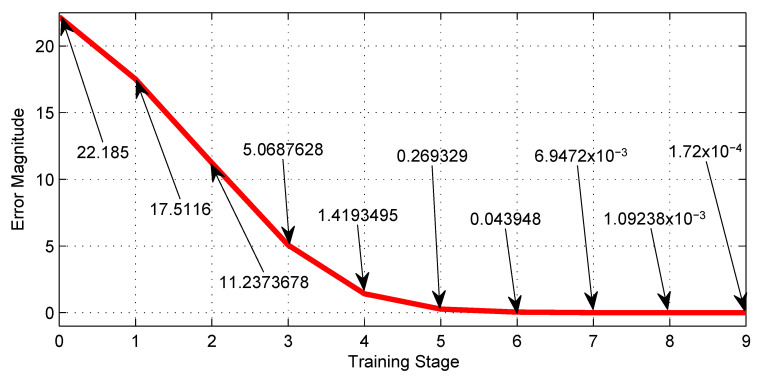
Error behavior of the NN with thirteen neurons given by Figure 9 along the training stages from the initial 0 to the 9th.

**Figure 21 sensors-23-08347-f021:**
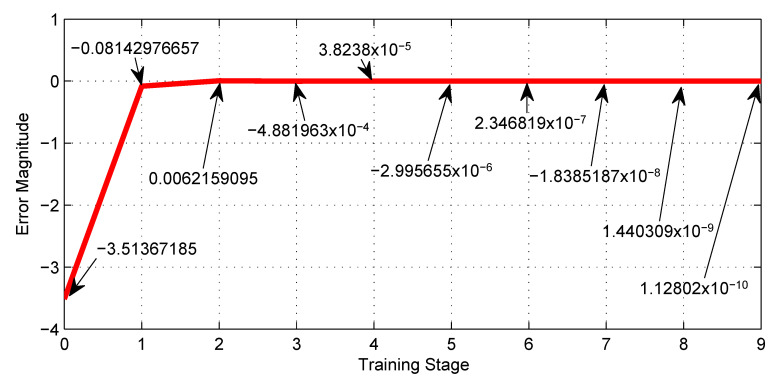
Error behavior of the NN with thirty-six neurons given by Figure 10 along the training stages from the initial 0 to the 9th.

**Figure 22 sensors-23-08347-f022:**
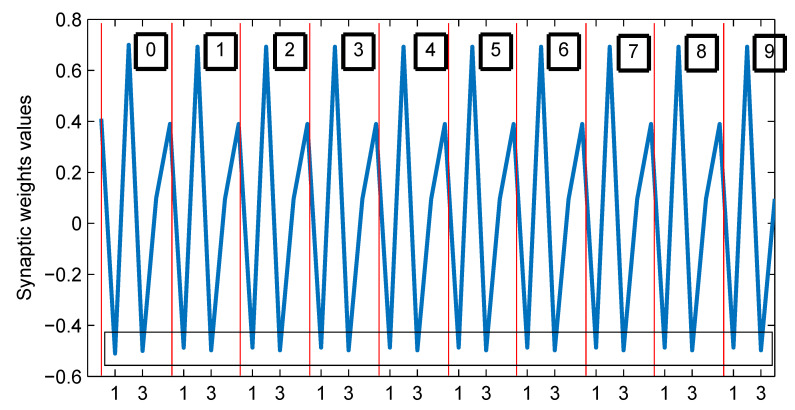
Behavior of the synaptic weights values of $w_{1}$ from 0 to the 9th training stage.

**Figure 23 sensors-23-08347-f023:**
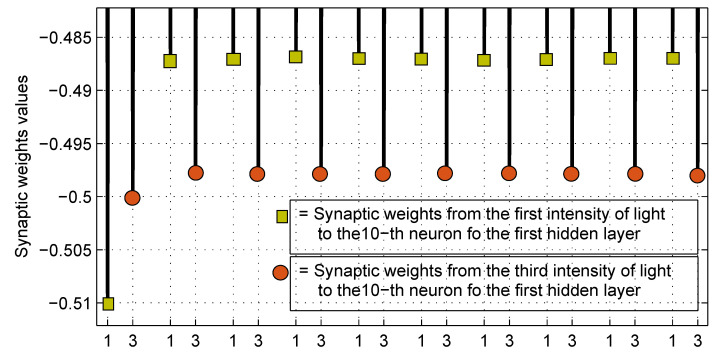
Enlarged view into the rectangle at the bottom of Figure 22.

**Table 1 sensors-23-08347-t001:** Minimum error in training for angular intervals.

	NN 2	NN 3	NN 4
**Interval**	**9 Neurons**	**13 Neurons**	**36 Neurons**
16-17-18	0.007742 (LR = $2.705\times 10^{-4}$)	0.0029 (LR = $1.29\times 10^{-4}$)	0.00474 (LR = $5.35\times 10^{-4}$)
15-17-19	0.01664 (LR = $2.6\times 10^{-4}$)	−0.08481 (LR = $8.5\times 10^{-5}$)	−0.01136 (LR = $3.9\times 10^{-4}$)
14-17-20	−0.03545 (LR = $2.5\times 10^{-4}$)	0.08497 (LR = $6\times 10^{-5}$)	0.02585 (LR = $3\times 10^{-4}$)
13-17-21	−0.4323 (LR = $2\times 10^{-4}$)	−0.8673 (LR = $5\times 10^{-5}$)	−0.303 (LR = $3\times 10^{-4}$)

**Table 2 sensors-23-08347-t002:** Error magnitude of the NN with three neurons of Figure 7, taking different learning rates (LR), along the nine training stages, where the best LR is 0.0005 (in bold).

LR	0	1	2	3	4	5	6	7	8	9
0.003	7.24	−56.8	2 $\times \;10^{4}$	−1 $\times \;10^{19}$	−4 $\times \;10^{102}$	nan	nan	nan	nan	nan
0.002	7.24	−33.08	225.1	−3 $\times \;10^{7}$	−2 $\times \;10^{36}$	2 $\times \;10^{199}$	nan	nan	nan	nan
0.001	7.24	−11.14	12.63	−9.032	4.085	−0.4246	0.065	−9.2 $\times \;10^{-3}$	1.3 $\times \;10^{-3}$	−1.9 $\times \;10^{-4}$
**0.0005**	7.24	−1.398	0.3467	−0.079	0.0184	−4.3 $\times \;10^{-3}$	9.9 $\times \;10^{-4}$	−2.3 $\times \;10^{-4}$	5.3 $\times \;10^{-5}$	−1.2 $\times \;10^{-5}$
0.0001	7.24	5.61	4.318	3.303	2.513	1.905	1.439	1.084	0.8156	0.6124
1 $\times \;10^{-5}$	7.24	7.077	6.92	6.766	6.615	6.457	6.321	6.179	6.039	5.903

**Table 3 sensors-23-08347-t003:** Error magnitude of the NN with nine neurons of Figure 8, taking different learning rates (LR), along the nine training stages, where the best LR is 0.0001 (in bold).

LR	0	1	2	3	4	5	6	7	8	9
0.003	−23.9	−6.6 $\times \;10^{4}$	−3.7 $\times \;10^{50}$	nan	nan	nan	nan	nan	nan	nan
0.002	−23.9	−1.3 $\times \;10^{4}$	−6 $\times \;10^{38}$	nan	nan	nan	nan	nan	nan	nan
0.001	−23.9	−657.5	−1.8 $\times \;10^{20}$	−4 $\times \;10^{250}$	nan	nan	nan	nan	nan	nan
0.0005	−23.9	−48.67	−5170	5 $\times \;10^{30}$	nan	nan	nan	nan	nan	nan
**0.0001**	−23.9	−8.187	−5.309	−3.693	−2.638	−1.905	−1.383	−1.006	−0.732	−0.533
1 $\times \;10^{-5}$	−23.9	−21.59	−19.71	−18.11	−16.75	−15.56	−14.53	−13.6	−12.78	−12.04

**Table 4 sensors-23-08347-t004:** Error magnitude of the NN with thirteen neurons of Figure 9, taking different learning rates (LR), along the nine training stages, where the best LR is 0.0001 (in bold).

LR	0	1	2	3	4	5	6	7	8	9
0.003	22.19	−1 $\times \;10^{4}$	4 $\times \;10^{38}$	nan	nan	nan	nan	nan	nan	nan
0.002	22.19	−2380	4 $\times \;10^{28}$	nan	nan	nan	nan	nan	nan	nan
0.001	22.19	−271.5	2 $\times \;10^{14}$	−1 $\times \;10^{171}$	nan	nan	nan	nan	nan	nan
0.0005	22.19	−37.81	−254.6	−2 $\times \;10^{12}$	−3 $\times \;10^{143}$	nan	nan	nan	nan	nan
**0.0001**	22.19	17.51	11.24	5.069	1.42	0.2693	0.044	6.9 $\times \;10^{-3}$	1.1 $\times \;10^{-3}$	1.7 $\times \;10^{-4}$
1 $\times \;10^{-5}$	22.19	21.83	21.46	21.08	20.67	20.25	19.82	19.37	18.9	18.42

**Table 5 sensors-23-08347-t005:** Error magnitude of the NN with thirty-six neurons of Figure 10, taking different learning rates (LR), along the nine training stages, where the best LR is 0.001 (in bold).

LR	0	1	2	3	4	5	6	7	8	9
0.003	3.51	3.19	−8.635	13.15	$2.4\times 10^{4}$	$5.6\times 10^{41}$	nan	nan	nan	nan
0.002	3.51	1.986	−2.851	1.13	0.693	0.296	−0.1436	0.06515	0.0304	0.014
**0.001**	3.51	−0.081	0.0062	$4.9\times 10^{-4}$	$3.82\times 10^{-5}$	−2.99 $\times \;10^{-6}$	$2.35\times 10^{-7}$	−1.84 $\times \;10^{-8}$	1.44 $\times \;10^{-9}$	−1.1 $\times \;10^{-10}$
0.0005	3.51	−1.59	0.7184	−0.3263	0.1487	0.06788	0.031	0.01418	6.48 $\times \;10^{-3}$	2.96 $\times \;10^{-3}$
0.0001	3.51	−3.09	−2.72	−2.398	2.116	1.869	1.652	−1.462	−1.294	−1.147
1 $\times \;10^{-5}$	3.51	−3.47	−3.427	−3.385	−3.344	−3.303	−3.262	−3.222	−3.183	−3.144

**Table 6 sensors-23-08347-t006:** Intensities of light with $I_{0}=50$.

Angle (Deg)	$I_{1}$	$I_{2}$	$I_{3}$	$I_{4}$
0	50	25	0	25
11.25	48.1	15.42	1.9	34.57
22.5	42.7	7.32	7.32	42.7
33.75	34.57	1.9	15.43	48.1
45	25	0	25	50
56.25	15.43	1.9	34.57	48.1
67.5	7.32	7.32	42.7	42.7
78.75	1.9	15.43	48.1	34.57
90	0	25	50	25
101.25	1.9	34.57	48.1	15.43
112.5	7.32	42.7	42.7	7.32
123.75	15.43	48.1	34.57	1.9
135	25	50	25	0
146.25	34.57	48.1	15.43	1.9
157.5	42.7	42.7	7.32	7.32
168.75	48.1	34.57	1.9	15.43
180	50	25	0	25

## Data Availability

Not applicable.

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
