# Peer review of "Artificial Neural Networks with Machine Learning Design for a Polyphasic Encoder"

_sensors, 2023, doi:10.3390/s23208347_

Round 1
Reviewer 1 Report
1- The research design:
§ Several references less than 5 years old should be added, because most of the references are outdated.
§ Your machine learning technique must be explained with more details.
§ Add a paragraph to present some future works.
2- Language quality and text format: several improvements must be made for an acceptable English quality of this paper such as:
§ Line 2: Replace the definite article before “study” with “a”
§ Line 8: The verb “is” appears to be in the incorrect tense; revise it to “was”
§ Line 70: Consider deleting “in order” in this context.
§ Many other necessary improvements were noted in the uploaded file.
Author Response
Authors want to express they’re gratitude for the time and effort made by the Reviewer to help us in improving this manuscript in both the research design and on the quality of English Language. Thank you very much.
Response to Reviewer 1 Comments
Point 1. The research design:
"Several references less than 5 years old should be added, because most of the references are outdated."
We have made an effort to find the works that can provide the most relevant information for the development of this paper. Please see the bibliography included, highlighted in magenta.
"Your machine learning technique must be explained with more details."
More relevant details have been added to explain the machine learning technique.
"Add a paragraph to present some future Works."
At the end of the manuscript a paragraph to present some future Works has been added.
Point 2. Comments on the quality of English Language:
Language quality and text format: several improvements must be made for an aceptable English quality of this paper such as:
Line 2: Replace the definite article before “study” with “a”.
Line 8: The verb “is” appears to be in the incorrect tense; revise it to “was”.
Line 70: Consider deleting “in order” in this context.
A complete revision on the quality of English Language has been performed. Also lines 2, 8, 70 have been improved as suggested.
Additionally, the background in the introduction section and the references have been improved. Please see the text in magenta colour.
Please see the attachment.

Reviewer 2 Report
The authors used machine learning to estimate the angular position of polarizers based on received light intensities. They argued that their artificial neural network design could produce accurate results. However, the motivation of the study needs to be improved, and there are severe flaws in the manuscript. Equation (2) has clearly shown that the angular position can be calculated mathematically once input light intensities I1, I2, I3, and I4 are available. Even if the authors argue that input light intensities are inaccurate due to instrument uncertainties, the equation is the best way to estimate the angular position. Machine learning is also susceptible to input uncertainties. Machine learning can just be used to estimate the results calculated by Equation (2) and thus can’t produce results better than those from Equation (2). If so, why should we apply machine learning here? Moreover, the authors use mathematical equations (3)-(6) to generate samples, which is more like homework but not a research study.
Some minor comments
Lines 20-21: Please rewrite this sentence. I don’t understand its meaning.
Lines 38-42: This type of citation of previous work is inappropriate. Please incorporate references into sentences. For example, for the sentence in Lines 16-17, you can insert some citations after the sentence.
Lines 45-46: Similar to the above comment. How about “In particular, Alvarez-Rodriguez and Alicala Ochoa [16] proposed a novel optical encoder based on a phase shifting algorithm that utilizes the polarization properties of light”?
Line 48: Could you provide more details about “multiple factors”?
Line 59: delete “this study”?
Line 67: What do you mean by a machine learning technique in the learning stage?
Lines 91-96: Are you sure your understanding of the Malus’ law is correct? What do you mean by q is the angular position of the polarizer?
Line 161: “firs” to “first”
Equation 14: Why didn’t you use relative error but absolute error?
Line 221: What is also used (16)?
The language of the manuscript needs improvement, such as incomplete sentences and inappropriate usage of words, although I can understand what the authors want to say.
Author Response
First and foremost, the authors greatly appreciate the Reviewer for their valuable feedback, which helps us improve both the presentation and the content of this technical report. We also want to express our gratitude for the time the Reviewer has invested in conducting a detailed and thorough analysis, uncovering details that had eluded by the authors during the writing process. Thank you very much.
Response to Reviewer 2 Comments
Point 1. Comment.
The authors used machine learning to estimate the angular position of polarizers based on received light intensities. They argued that their artificial neural network design could produce accurate results. However, the motivation of the study needs to be improved, and there are sever flaws in the manuscript. Equation (2) has clearly shown that the angular position can be calculated mathematically once input light intensities l1, l2, l3 and l4 are available. Even if the authors argue that input light intensities are inaccurate due to instrument uncertainties, the equation is the best way to estimate the angular position. Machine learning is also susceptible to input uncertainties. Machine learning can just be used to estimate the results calculated by Equation (2) and thus can’t produce results better than those form Equation (2). If so, why should we apply machine learning here? Moreover, the authors use mathematical equations (3)— (6) to generate samples, which is more like homework but not a research study.
The reviewer is correct in pointing out that the advantages of using the neural network have not been clearly established. Therefore, we have included in both the problem setting section and the conclusions the reasons why the neural network provides a more accurate reading than the mathematical baseline algorithm. We had also omitted mentioning, in the initial version of the article, the slightly perturbed conditions in the simulation processes, as well as the modification of the original light intensity equations, to which a perturbation and noise term must be added to simulate a real environment during the training stage, in order for the machine learning technique to be effective. All of this has been corrected. Please see the changes highlighted in cyan color.
Point 2. Some minor comments.
Lines 20—21: Please rewrite this sentence. I don’t know its meaning.
The correction has been made. Please see the new paragraph.
Lines 38—42: This type of citation of previous work is inappropriate. Please incorpórate references into sentences. For example, for the sentence in lines 16—17, you can insert some citations after the sentence.
The correction has been made. Please see the new paragraph with text in cyan color.
Lines 45—46: Similar to the above comment. How about “In particular, Alvarez-Rodríguez and Alcalá Ochoa [16] proposed a novel optical encoder base on a phase shifting algorithm that utilizes the polarization properties of light”?
The correction has been made. Please see the text in cyan color.
Line 48: Could you provide more details about “multiple factors”?
The authors have added further details. Please refer to the text in cyan color.
Line 59: Delete “this study”?
Thank you very much for that observation, the correction has been made.
Line 67: What do you mean by a machine learning technique in the learning stage?
Thank you very much for that observation, the correction has been made.
Lines 91—96: Are you sure your understanding of the Malus’ law is correct? What do you mean by q is the angular position of the polarizer?
The redaction has been improved, please see the new paragraph in blue colour.
Line 161: “fist” to “first”.
Thank you very much for that observation, the correction has been made.
Equation 14: Why didn’t you use relative error but absolute error?
In text has been improved and has been added “the value we wish to minimize”.
Line 221: What is also used (16)?
The sentence has been improved.
Please see the attachment.

Reviewer 3 Report
The abstract presents a study on the application of artificial neural networks (ANNs) in information processing using an optical encoder based on the polarization of light. The paper proposes a machine learning technique to train neural networks to accurately predict the angular position of a rotating element in the neuro-encoder. Using light's phase shifting arrangements as input data enables the system to predict the angular position accurately.
The strength of this paper lies in its exploration of artificial neural networks in managing complex and difficult-to-interpret data, specifically in the context of optical encoders. The proposed neural designs demonstrate excellent performance, particularly in small angular intervals, a significant advantage for precise measurements.
One noteworthy aspect is the methodology proposed to retain the remarkable characteristic of the neural networks in measurements ranging from 0 to 180 degrees or even to 360 degrees. This approach is crucial, ensuring the neuro-encoder's performance remains consistent and reliable across various angular positions.
However, the abstract needs more specific details about the methodology for training the neural networks and the implementation of the simulation stage. It would be beneficial to include more information about the neural network architecture, the dataset used for training, and the evaluation metrics employed to assess the performance.
Moreover, the abstract mentions the potential applications of this study in improving the capabilities of resolvers or other polyphasic sensors. While this is an interesting prospect, it would be helpful to elaborate on the practical implications and potential benefits of integrating these neural designs into such sensor systems. I advise authors to include this work in their background https://doi.org/10.3390/app13020677In conclusion, the abstract presents an intriguing study on the application of artificial neural networks in an optical encoder based on light polarization. The emphasis on accuracy and the proposed methodology to maintain performance across various angular positions are commendable. However, the abstract could be further improved by providing more specific details about the methodology and discussing the practical implications of the findings in real-world sensor systems.
Author Response
According to your comments, the authors believe that the Reviewer finds this work useful, for which we are very thankful. Likewise, we would like to express our gratitude for your time and expertise in helping us improve the presentation of this manuscript.
Response to Reviewer 3 Comments
Point 1. The abstract presents a study on the application of artificial neural networks (ANNs) in information processing using an optical encoder based on the polarization of light. The paper proposes a machine learning technique to train neural networks to accurately predict the angular position of a rotating element in the neuro-encoder. Using light’s phase shifting arrangements as input data enables the system to predict the angular position accurately.
The strength of this paper lies in its exploration of artificial neural networks in managing complex and difficult-to-interpret data, specifically in the context of optical encoders. The proposed neural designs demonstrate excellent performance, particularly in small angular intervals, a significant advantage for precise mesurements.
One noteworthy aspect is the methodology proposed to retain the remarkable characteristic of the neural networks in measurements ranging from 0 to 180 degrees or even to 360 degrees. This approach is crucial, ensuring the neuro-encoder’s performance remains consistent and reliable across various angular positions.
However, the abstract needs more specific details about the methodology for training the neural networks and the implementation of the simulation stage. It would be beneficial to include more information about the neural network architecture, the dataset used for training, and the evaluation metrics employed to assess the performance.
The abstract has been rearranged to mention more information about the neural network architecture, and the evaluation metric employed. Also it is mentioned that the neural network is retropropagated (for training purposes). Please see the new abstract in cyan colour. The dataset used for training can be seen in the Apendix.
Point 2. Moreover, the abstract mentions the potential applications of this study in improving the capabilities of resolvers or other polyphasic sensors. While this is an interesting prospect, it would be helpful to elaborate on the practical implications and potential benefits of integrating these neural designs into such sensor systems. I advise authors to include this work in their background: doi.org/10.3390/app13020677.
Thank you very much for your suggestion, the proposed work has been included in the background of the paper. Please see the paragraph in cyan letter.
Point 3. In conclusion, the abstract presents an intriguing study on the application of artificial neural networks commendable. However, the abstract could be further improved by providing more specific details about the methodology and discussing the practical implications of the findings in real-world sensor systems.
The wording of the abstract has been modified. The authors hope that this new abstract provides the relevant information requested by the Reviewer, thank you very much.
Please see the attachment.

Round 2
Reviewer 1 Report
- Reduce number of oudateted references
- reconsider the format of some paragraphs in Appendix section to avoid text and line number clash.
Quality of English is acceptable.
Author Response
Dear Editor,
Beforehand, we are deeply grateful to you and to the Reviewers for your efforts and time spent in this work.
The Authors believe that each of the issues and comments from the Reviewers have been addressed in an exhaustive and satisfactory manner, which have undoubtedly helped us to improve both the content and the presentation of the manuscript.
Response to Reviewer 1.
Comment 1:
Reduce number of outdated references.
Response 1: Thank you for pointing this out. Three references which are out of date were eliminated from both the background and References section.
Comment 2:
Reconsider the format of some paragraphs in Appendix section to avoid text and line number clash.
Response 2: We agree with this comment. Therefore, the format of Appendix has been modified to clarify the information contained therein. The Appendix has been divided in four parts (A, B, C, D). Please see adecuations in magenta color inside the manuscript.
We are deeply grateful to Reviewer 1 for his invaluable help in improving this work.

Reviewer 2 Report
The authors addressed some of my questions. However, the critical issue is the need for more motivation to apply machine learning to the problem introduced in the study. I understand that the mathematical equations are affected by measurement uncertainties, which similarly affect your machine learning results. In reality, we know the measurements are inaccurate, but we have no idea how much the measurement uncertainties are (If we know, we can solve the problem explicitly using those mathematical equations). If so, how can you know machine learning is better than those mathematical equations? I cannot recommend the publication of the manuscript.
Author Response
Dear Editor,
Beforehand, we are deeply grateful to you and to the Reviewers for your efforts and time spent in this work.
The Authors believe that each of the issues and comments from the Reviewers have been addressed in an exhaustive and satisfactory manner, which have undoubtedly helped us to improve both the content and the presentation of the manuscript.
Reviewer 2 has expressed his concerns regarding the self-learning technique, although this concern has been addressed, we can additionally establish that:
- Every experiment and every result given herein, is reproducible by any reader in any computer with Matlab-Simulink installed (Section: Implementation example).
- Figure 14 shows the block diagram as an implementation example of the machine learning technique.
- Figures 18–23, along with Tables 1—5, are devoted to show the outstanding peformance of the machine learning technique. If more than these 6 figures and 5 tables, to show the performance of the machine learning are included in the manuscript, then, this technical report will be much more extensive, see that the paper currently has 26 pages. The minimum, necessary and sufficient amount of material has been included.
- The math-based machine learning technique has been modelled by equations (20)—(25), including static and dynamic disturbances for the real-world.
The comment of the Reviewer 2 is noted below and the action taken is detailed in response.
Response to Reviewer 2 Comments
Comment 1: The authors addressed some of my questions. However, the critical issue is the need for more motivation to apply machine learning to the problem introduced in the study. I understand that the mathematical equations are affected by measurement uncertainties, which similarly affect your machine learning results. In reality, we know the measurements are inaccurate, but we have no idea how much the measurement uncertainties are (If we know, we can solve the problem explicitly using those mathematical equations). If so, how can you know machine learning is better than those mathematical equations? I cannot recommend the publication of the manuscript.
Response 1: The Reviewer is correct in asking us to clarify the motivation for the use of the self-learning technique, since this strategy allowed us to bring this work to a successful conclusion, saving a great deal of time and effort in obtaining results and in training the neural networks. Likewise, the Authors are convinced that this innovative technique based on math-modelling will be very useful for other people, just as it has been for us.
To highlight the motivation in using the machine learning technique, in a previous version the following paragraph says in subsection 3.3 that “The acquisition of information vectors for training a neural network is a time-consuming and laborious process. For this reason, it is desirable for the machine to have the ability to learn by itself using some strategy”. To reinforce the last sentence, in the same subsection the following paragraph has been included (in blue): “The another alternative to not using the machine learning technique is to train the network manually, which would increase in an extreme way the time and effort devoted to obtain results and adjustments of the neural networks”.
Other paragraphs to clarify the benefits in using the machine learning technique, have been included (in blue):
In section 4. “The initial readings in column 0 were obtained using randomly assigned synaptic weights, while the machine learning technique updated the synaptic weights from the 1-th to the 9-th stages in an extremely fast process.”, and, “Obtaining the results presented in each of the above tables using manual neural training methods can take several days of hard work, per each run, while the use of the proposed computer machine learning technique has reduced that time to only a few minutes, per each run, using a standard computer.”
In Conclusions. “The most notable advantage in using the machine learning technique, exemplified by Figure 14, is the enormous savings in time and effort in tuning synaptic weights for the neural networks. Even more, Figures 18--21 show the outstanding performance of the proposed math-based self-learning strategy.”
In Future Work. “Another future work can be a deeper study of the math-based machine learning technique, focused on modelling both the static and the dynamic disturbances $\lambda_k$ for every kind of polyphasic sensor.”
We also fully agree with the reviewer that this novel self-learning technique is the reason for future work that deepens its study and application in all types of polyphase sensors. In the current work, enough information is offered for its implementation, and results of its performance are given in many Figures and Tables, even though this manuscript has already extended to 26 pages (which is more than recommended).
The Authors thank Reviewer 2 for his time and effort spent in helping to improve this manuscript, and we believe that each of your comments has been rigorously and carefully addressed.
Thank you very much.
